# GAN2GAN: Generative Noise Learning for Blind Denoising with Single Noisy Images

**Sungmin Cha[1], Taeeon Park[1], Byeongjoon Kim[2], Jongduk Baek[2] and Taesup Moon[3]***
Sungkyunkwan University[1], Yonsei University[2], Seoul National University[3], South Korea
`{csm9493,pte1236}@skku.edu, bjkim2006@naver.com,`
`jongdukbaek@yonsei.ac.kr, tsmoon@snu.ac.kr`

## Abstract

We tackle a challenging blind image denoising problem, in which only single distinct noisy images are available for training a denoiser, and no information about noise is known, except for it being zero-mean, additive, and independent of the clean image. In such a setting, which often occurs in practice, it is not possible to train a denoiser with the standard discriminative training or with the recently developed Noise2Noise (N2N) training; the former requires the underlying clean image for the given noisy image, and the latter requires two independently realized noisy image pair for a clean image. To that end, we propose GAN2GAN (Generated-Artificial-Noise to Generated-Artificial-Noise) method that first learns a generative model that can 1) simulate the noise in the given noisy images and 2) generate a rough, noisy estimates of the clean images, then 3) iteratively trains a denoiser with subsequently synthesized noisy image pairs (as in N2N), obtained from the generative model. In results, we show the denoiser trained with our GAN2GAN achieves an impressive denoising performance on both synthetic and real-world datasets for the blind denoising setting; it almost approaches the performance of the standard discriminatively-trained or N2N-trained models that have more information than ours, and it significantly outperforms the recent baseline for the same setting, *e.g.*, Noise2Void, and a more conventional yet strong one, BM3D. The official code of our method is available at https://github.com/csm9493/GAN2GAN.

## 1 Introduction

Image denoising is one of the oldest problems in image processing and low-level computer vision, yet it still attracts lots of attention due to the fundamental nature of the problem. A vast number of algorithms have been proposed over the past several decades, and recently, the CNN-based methods, *e.g.*, Cha & Moon (2019); Zhang et al. (2017); Tai et al. (2017); Liu et al. (2018), became the throne-holders in terms of the PSNR performance. The main approach of the most CNN-based denoisers is to apply the discriminative learning framework with (clean, noisy) image pairs and *known* noise distribution assumption. While being effective, such framework also possesses a couple of limitations that become critical in practice; the assumed noise distribution may be mismatched to the actual noise in the data or obtaining the noise-free clean target images is not always possible or very expensive, *e.g.*, medical imaging (CT or MRI) or astrophotographs.

Several attempts have been made to resolve above issues. For the noise uncertainty, the so-called *blind training* have been proposed. Namely, a denoiser can be trained with a composite training set that contains images corrupted with multiple, pre-defined noise levels or distributions, and such blindly trained denoisers, *e.g.*, DnCNN-B in Zhang et al. (2017), were shown to alleviate the mismatch scenarios to some extent. However, the second limitation, *i.e.*, the requirement of clean images for building the training set, still remains. As an attempt to address this second limitation, Lehtinen et al. (2018) recently proposed the Noise2Noise (N2N) method. It has been shown that a denoiser, which has a negligible performance loss, can be trained without the clean target images, as long as two independent noisy image realizations for the same underlying clean image are available. Despite its

---

*Corresponding author (E-mail: `tsmoon@snu.ac.kr`)

effectiveness, the requirement of the *two* independently realized noisy image pair for a single clean image, which may hardly be available in practice, is a critical limiting factor for N2N.

In this paper, we consider a setting in which neither of above approach is applicable, namely, the pure *unsupervised* blind denoising setting where only *single* distinct noisy images are available for training. Namely, nothing is known about the noise other than it being zero-mean, additive, and independent of the clean image, and neither the clean target images for blind training nor the noisy image pairs for N2N training is available. While some recent work, *e.g.*, Krull et al. (2019); Batson & Royer (2019); Laine et al. (2019), took the self-supervised learning (SSL) approach for the same setting, we take a generative learning approach. The crux of our method is to first learn a Wasserstein GAN (Arjovsky et al., 2017)-based generative model that can 1) learn and simulate the noise in the given noisy images and 2) generate rough, initially denoised images. Using such generative model, we then synthesize noisy image pairs by corrupting each of the initially denoised images with the simulated noise *twice* and use them to train a CNN denoiser as in the N2N training (*i.e.*, Noisy N2N). We further show that *iterative* N2N training with refined denoised images can significantly improve the final denoising performance. We dubbed our method as GAN2GAN (Generated-Artifical-Noise to Generated-Artificial-Noise) and show that the denoiser trained with our method can achieve (sometimes, even outperform) the performance of the standard supervised-trained or N2N-trained blind denoisers for the white Gaussian noise case. Furthermore, for mixture/correlated noise or real-world noise in microscopy/CT images, for which the exact distributions are hard to know *a priori*, we show our denoiser significantly outperforms those standard blind denoisers, which are mismatch-trained with white Gaussian noise, as well as other baselines that operate in the same condition as ours: the SSL baseline, N2V (Krull et al., 2019), and a more conventional BM3D (Dabov et al., 2007).

## 2 RELATED WORK

Several works have been proposed to overcome the limitation of the vanilla supervised learning based denoising. As mentioned above, Noise2Self (N2S) (Batson & Royer, 2019) and Noise2Void (N2V) (Krull et al., 2019) recently applied self-supervised learning (SSL) approach to train a denoiser only with single noisy images. Their settings exactly coincide with ours, but we show later that our GAN2GAN significantly outperforms them. More recently, Laine et al. (2019) improved N2V by incorporating specific noise likelihood models with Bayesian framework, however, their method *required* to know the exact noise model and could not be applied to more general, unknown noise settings. Similarly, Soltanayev & Chun (2018) proposed SURE (Stein's Unbiased Risk Estimator)-based denoiser that can also be trained with single noisy images, but it worked only with the *Gaussian* noise. Their work was extended in Zhussip et al. (2019), but it required noisy image *pairs* as in N2N as well as the Gaussian noise constraint. Chen et al. (2018) devised GCBD method to learn and generate noise in the given noisy images using W-GAN Arjovsky et al. (2017) and utilized the unpaired clean images to build a supervised training set. Our GAN2GAN is related to Chen et al. (2018), but we significantly improve their noise learning step and do *not* use the clean data at all. Table 1 summarizes and compares the settings among the above mentioned recent baselines. We clearly see that only our GAN2GAN and N2V do not utilize any "sidekicks" that other methods use.

Table 1: Summary of different settings among the recent baselines.

| Alg.\ Requirements | Clean image | Noisy "pairs" | Noise model |
|---|---|---|---|
| N2N [Lehtinen et al. (2018)] | ✗ | ✓ | ✗ |
| HQ SSL [Laine et al. (2019)] | ✗ | ✗ | ✓ |
| SURE [Soltanayev & Chun (2018)] | ✗ | ✗ | ✓ |
| Ext. SURE [Zhussip et al. (2019)] | ✗ | ✓ | ✓ |
| GCBD [Chen et al. (2018)] | ✓ | ✗ | ✗ |
| N2V [Krull et al. (2019)] | ✗ | ✗ | ✗ |
| GAN2GAN (Ours) | ✗ | ✗ | ✗ |

Additionally, there are recently published papers on blind image denoising but these also have a difference with ours. Anwar & Barnes (2019); Zhang et al. (2018) suggest effective CNN architectures for denoising, however, they only consider the setting in which clean images are necessary for training. Zamir et al. (2020) considers the denoising of specific camera settings, and it also requires clean sRGB images as well as the knowledge of the noise level. Thus, it cannot be applied to the complete blind setting as ours, in which no information on the specific noise distribution or clean images is available.

More classical denoising methods are capable of denoising solely based on the single noisy images by applying various principles, *e.g.*, filtering-based Buades et al. (2005); Dabov et al. (2007), optimization-based Elad & Aharon (2006); Mairal et al. (2009), Wavelet-based Donoho & Johnstone (1995), and effective prior-based Zoran & Weiss (2011). Those methods typically are, however, computationally intensive during the inference time and cannot be *trained* from a separate set of noisy images, which limits their denoising performance. Another line of recent work worth mentioning is the deep learning-based priors or regularizers, *e.g.*, Ulyanov et al. (2018); Yeh et al. (2018); Lunz et al. (2018), but their PSNRs still fell short of the supervised trained CNN-based denoisers.

## 3 MOTIVATION

In order to develop the core intuition for motivating our method, we first consider a simple, single-letter Gaussian noise setting. Let $Z = X + N$ be the noisy observation of $X \sim \mathcal{N}(0, \sigma_X^2)$, corrupted by the $N \sim \mathcal{N}(0, \sigma_N^2)$. It is well known that the minimum MSE (MMSE) estimator of $X$ given $Z$ is $f_{\text{MMSE}}^*(Z) = \mathbb{E}(X|Z) = \frac{\sigma_X^2}{\sigma_X^2 + \sigma_N^2} Z$. We now identify the optimality of N2N in this setting.

**N2N** Assume that we have two i.i.d. copies of the noise $N$: $N_1$ and $N_2$. Then, let $Z_1 = X + N_1$ and $Z_2 = X + N_2$ be the two independent noisy observation pairs of $X$. The N2N in this setting corresponds to obtaining the MMSE estimator of $Z_2$ given $Z_1$,

$$f_{\text{N2N}}(Z_1) \triangleq \arg\min_f \mathbb{E}(Z_2 - f(Z_1))^2 = \mathbb{E}(Z_2|Z_1) = \mathbb{E}(X + N_2|Z_1) \overset{(a)}{=} \mathbb{E}(X|Z_1) = \frac{\sigma_X^2}{\sigma_X^2 + \sigma_N^2} Z_1, \quad (1)$$

in which (a) follows from $N_2$ being independent of $Z_1$. Note (1) has the exact same form as $f_{\text{MMSE}}^*(Z)$, hence, estimating $X$ with $f_{\text{N2N}}(Z)$ also achieves the MMSE, in line with (Lehtinen et al., 2018).

**"Noisy" N2N** Now, consider the case in which we again have the two i.i.d. $N_1$ and $N_2$, but the noisy observations are of a *noisy* version of $X$. Namely, let $X' = X + N_0$, in which $N_0 \sim \mathcal{N}(0, \sigma_0^2)$, and denote $Z_1' = X' + N_1$ and $Z_2' = X' + N_2$ as the noisy observation pairs. Then, we can define a "Noisy" N2N estimator as the MMSE estimator of $Z_2'$ given $Z_1'$,

$$f_{\text{Noisy N2N}}(Z_1', y) \triangleq \arg\min_f \mathbb{E}(Z_2' - f(Z_1'))^2 = \mathbb{E}(X'|Z_1') = \frac{\sigma_X^2(1+y)}{\sigma_X^2(1+y) + \sigma_N^2} Z_1', \quad (2)$$

in which we denote $y \triangleq \sigma_0^2/\sigma_X^2$ and assume that $0 \le y < 1$. Note clearly (2) coincides with (1) when $y = \sigma_0^2 = 0$. Following N2N, (2) is essentially estimating $X'$ based on $Z' = X' + N$. An interesting subtle question is what happens when we use the mapping $f_{\text{Noisy N2N}}(Z, y)$ for estimating $X$ given $Z = X + N$, *not* $X'$ given $Z'$. Our theorem below, of which proof is in the Supplementary Material (S.M.), shows that for a sufficiently large $\sigma_0^2$, $f_{\text{Noisy N2N}}(Z, y)$ gives a better estimate of $X$ than $X'$.

**Theorem 1** *Consider the single-letter Gaussian setting and $f_{\text{Noisy N2N}}(Z, y)$ obtained in (2). Also, assume $0 < y < 1$. Then, there exists some $y_0$ s.t. $\forall y \in (y_0, 1)$, $\mathbb{E}(X - f_{\text{Noisy N2N}}(Z, y))^2 < \sigma_0^2$.*

Theorem 1 provides a simple, but useful, intuition that motivates our method; if simulating the noise in the images is possible, we may carry out the N2N training iteratively, provided that a rough *noisy* estimate of the clean image is initially available. Namely, we can first simulate the noise to generate noisy observation pairs of the initial noisy estimate, then do the Noisy N2N training with them to obtain a denoiser that may result in a better estimate of the clean image when applied to the actual noisy image subject to denoising (as in Theorem 1). Then, we can refine the estimates by *iterating* the Noisy N2N training with the generated noisy observation pairs of the previous step's estimate of the clean image, until convergence.

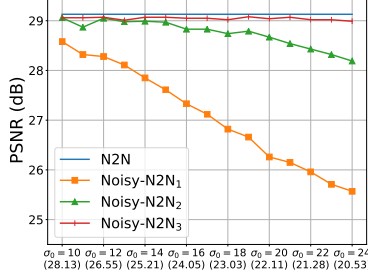

Figure 1: Iterative Noisy N2N.

To check whether above intuition is valid, we carry out a feasibility experiment. Figure 1 shows the denoising results on BSD68 (Roth & Black, 2009) for Gaussian noise with $\sigma = 25$. The blue line is the PSNR of the N2N

model trained with noisy observation pairs of the *clean* images in the BSD training set, serving as an upper bound. The orange line, in contrast, is the PSNR of the Noisy $N2N_1$ model that is trained with the noisy observation pairs of the *noisy* estimates for the clean images, which were set to be another Gaussian noise-corrupted training images. The standard deviations ($\sigma_0$) of the Gaussian for generating the noisy estimates are given in the horizontal axis, and the corresponding PSNRs of the estimates are given in the parentheses. Although Noisy $N2N_1$ clearly lies much lower than the N2N upper bound, we note its PSNR is still higher than that of the initial noisy estimates, which is in line with Theorem 1. Now, if we iterate the Noisy N2N with the previous step's denoised images (*i.e.*, Noisy-$N2N_2$/Noisy-$N2N_3$ for second/third iterations, respectively), we observe that the PSNR significantly improves and approaches the ordinary N2N for most of the initial $\sigma_0$ values. Thus, we observe the intuition from Theorem 1 generalizes well to the image denoising case in an ideal setting, where the noise can be perfectly simulated, and the initial noisy estimates are Gaussian corrupted versions. The remaining question is whether we can also obtain similar results for the blind image denoising setting. We show our generative model-based approach in details in the next section.

## 4 MAIN METHOD: THREE COMPONENTS OF GAN2GAN

To concretely describe our method, we first set the notations. We assume the noisy image $\mathbf{Z}$ is generated by $\mathbf{Z} = \mathbf{x} + \mathbf{N}$, in which $\mathbf{x}$ denotes the underlying clean image and $\mathbf{N}$ denotes the zero-mean, additive noise that is independent of $\mathbf{x}$. For training a denoiser, we do not assume either the distribution or the covariance of $\mathbf{N}$ is known. Moreover, we assume only a database of $n$ *distinct* noisy images, $\mathcal{D} = \{\mathbf{Z}^{(i)}\}_{i=1}^n$, is available for learning a denoiser. A CNN-based denoiser is denoted as $\hat{\mathbf{X}}_\phi(\mathbf{Z})$ with $\phi$ being the model parameter, and we use the standard quality metrics, PSNR/SSIM, for evaluation. Our method consists of three parts; 1) smooth noisy patch extraction, 2) training a generative model, and 3) iterative GAN2GAN training of $\hat{\mathbf{X}}_\phi(\mathbf{Z})$, each of which we elaborate below.

### 4.1 SMOOTH NOISY PATCH EXTRACTION

The first step is to extract the noisy image patches from $\mathcal{D}$ that correspond to smooth, homogeneous areas. Our extraction method is similar to that of the GCBD proposed in (Chen et al., 2018), but we make a critical improvement. The GCBD determines a patch $\boldsymbol{p}$ (of pre-determined size) is smooth if it satisfies the following for *all* of its smaller sub-patches, $\mathbf{q}_j$, with some hyperparameters $\mu, \gamma \in (0, 1)$:

$$|\mathbb{E}(\mathbf{q}_j) - \mathbb{E}(\boldsymbol{p})| \le \mu \mathbb{E}(\boldsymbol{p}), \ \ |\mathbb{V}(\mathbf{q}_j) - \mathbb{V}(\boldsymbol{p})| \le \gamma \mathbb{V}(\boldsymbol{p}), \tag{3}$$

in which $\mathbb{E}(\cdot)$ and $\mathbb{V}(\cdot)$ are the empirical mean and variance of the pixel values.

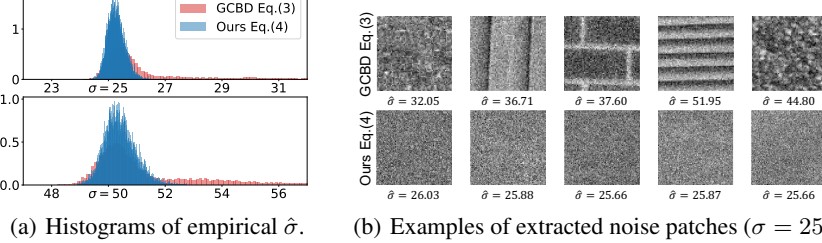

(a) Histograms of empirical $\hat{\sigma}$.     (b) Examples of extracted noise patches ($\sigma = 25$)

Figure 2: Comparison of smooth noisy patch extraction rules.

While (3) works for extracting smooth patches to some extent, as we show in Figure 2(b), it does not rule out choosing patches with high-frequency repeating patterns, which are far from being smooth. Thus, we instead use the 2D discrete wavelet transform (DWT) for a new extraction rule; namely, we determine $\boldsymbol{p}$ is smooth if its four sub-band decompositions obtained by DWT, $\{W_k(\boldsymbol{p})\}_{k=1}^4$, satisfy

$$\frac{1}{4} \sum_{k=1}^4 \left| \hat{\boldsymbol{\sigma}}(W_k(\boldsymbol{p})) - \mathbb{E}[\hat{\boldsymbol{\sigma}}_W(\boldsymbol{p})] \right| \le \lambda \mathbb{E}[\hat{\boldsymbol{\sigma}}_W(\boldsymbol{p})], \tag{4}$$

in which $\hat{\boldsymbol{\sigma}}(\cdot)$ is the empirical standard deviation of the wavelet coefficients, $\mathbb{E}[\hat{\boldsymbol{\sigma}}_W(\boldsymbol{p})] \triangleq \frac{1}{4} \sum_{k=1}^4 \hat{\boldsymbol{\sigma}}(W_k(\boldsymbol{p}))$, and $\lambda \in (0, 1)$ is a hyperparameter. This rule is much simpler than (3), which

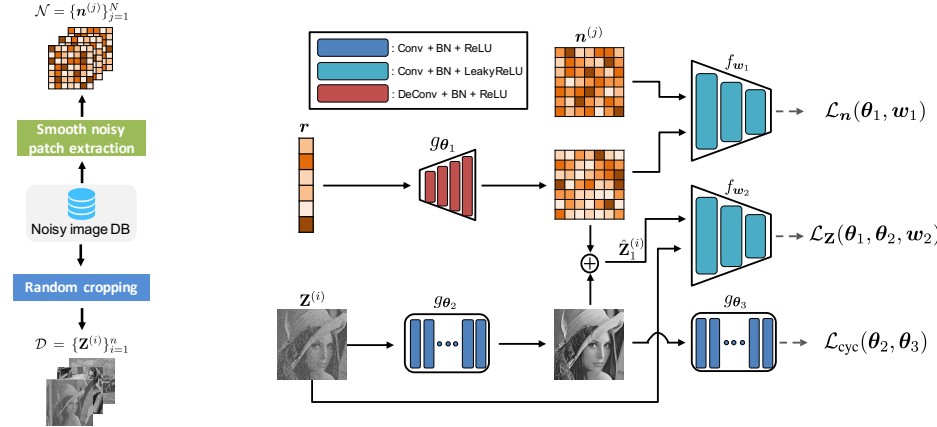

(a) Getting $\mathcal{D}$ & $\mathcal{N}$    (b) The model architecture with three generators and two critics.

Figure 3: Overall structure of the W-GAN based generative model.

has to be evaluated for all the sub-patches, $\{\mathbf{q}_j\}$. Once $N$ patches are extracted from $\mathcal{D}$ using (4), we subtract each patch with its mean pixel value, and obtain a set of "noise" patches, $\mathcal{N} = \{\boldsymbol{n}^{(j)}\}_{j=1}^N$. Such subtraction is valid since all the pixel values should be close to their mean in a smooth patch, and the noise is assumed to be zero-mean, additive.

Figure 2 compares the rules (3) and (4) by showing the quality of the "noise" patches extracted from 1,000 Gaussian-corrupted images . The two plots in Figure 2(a) show the normalized histograms of the empirical standard deviations, $\hat{\sigma}$, of the extracted patches when true $\sigma = \{25, 50\}$, respectively. We clearly observe that while the $\hat{\sigma}$'s for (4) are mostly concentrated on true $\sigma$, those of (3) have much higher variation. In addition, Figure 2(b) visualizes the randomly sampled patches of which $\hat{\sigma}$'s were above the 90-th percentile among the extracted patches for each rule (when $\sigma = 25$). Again, it is obvious that (3) also may result in selecting the patches with high-frequency patterns, whereas (4) is much more effective for extracting accurate noise patches. Later, we show (in Figure 5) that such improved quality of the noise patches by our (4) plays an *essential* role; namely, our pure unsupervised learning based denoiser using (4) even outperforms the clean target image based denoiser in (Chen et al., 2018) using (3).

## 4.2 TRAINING A W-GAN BASED GENERATIVE MODEL

Equipped with $\mathcal{D} = \{\mathbf{Z}^{(i)}\}_{i=1}^n$ and the extracted noise patches $\mathcal{N} = \{\boldsymbol{n}^{(j)}\}_{j=1}^N$, we train a generative model, which can learn and simulate the noise as well as generate initial noisy estimates of the clean images, hence, realize the Noisy N2N training explained in Section 3. As shown in Figure 3, our model has three generators, $\{g_{\boldsymbol{\theta}_1}, g_{\boldsymbol{\theta}_2}, g_{\boldsymbol{\theta}_3}\}$, and two critics, $\{f_{\boldsymbol{w}_1}, f_{\boldsymbol{w}_2}\}$, in which the subscripts stand for the model parameters. The loss functions associated with the components of our model are:

$$\mathcal{L}_{\boldsymbol{n}}(\boldsymbol{\theta}_1, \boldsymbol{w}_1) \triangleq \mathbb{E}_{\boldsymbol{n}}\big[f_{\boldsymbol{w}_1}(\boldsymbol{n})\big] - \mathbb{E}_{\boldsymbol{r}}[f_{\boldsymbol{w}_1}(g_{\boldsymbol{\theta}_1}(\boldsymbol{r}))] \tag{5}$$

$$\mathcal{L}_{\mathbf{Z}}(\boldsymbol{\theta}_1, \boldsymbol{\theta}_2, \boldsymbol{w}_2) \triangleq \mathbb{E}_{\mathbf{Z}}\big[f_{\boldsymbol{w}_2}(\mathbf{Z})\big] - \mathbb{E}_{\mathbf{Z}, \boldsymbol{r}}\big[f_{\boldsymbol{w}_2}(g_{\boldsymbol{\theta}_2}(\mathbf{Z}) + g_{\boldsymbol{\theta}_1}(\boldsymbol{r}))\big] \tag{6}$$

$$\mathcal{L}_{\mathrm{cyc}}(\boldsymbol{\theta}_2, \boldsymbol{\theta}_3) \triangleq \mathbb{E}_{\mathbf{Z}}\big[\|\boldsymbol{z} - g_{\boldsymbol{\theta}_3}(g_{\boldsymbol{\theta}_2}(\mathbf{Z}))\|_1\big]. \tag{7}$$

The loss (5) is a standard W-GAN (Arjovsky et al., 2017) loss for training the first generator-critic pair, $(g_{\boldsymbol{\theta}_1}, f_{\boldsymbol{w}_1})$, of which $g_{\boldsymbol{\theta}_1}$ learns to generate the independent realization of the noise mimicking the patches in $\mathcal{N} = \{\boldsymbol{n}^{(j)}\}_{j=1}^N$, taking the random vector $\boldsymbol{r} \sim \mathcal{N}(0, I)$ as input. The second loss (6) links the two generators, $g_{\boldsymbol{\theta}_1}$ and $g_{\boldsymbol{\theta}_2}$, with the second critic, $f_{\boldsymbol{w}_2}$. The second generator $g_{\boldsymbol{\theta}_2}$ is intended to generate the *estimate* of the underlying clean patch for $\mathbf{Z}$, *i.e.*, coarsely denoise $\mathbf{Z}$, and the critic $f_{\boldsymbol{w}_2}$ determines how close the distribution of the *generated* noisy image, $g_{\boldsymbol{\theta}_2}(\mathbf{Z}) + g_{\boldsymbol{\theta}_1}(\boldsymbol{r})$, is to the that of $\mathbf{Z}$[1]. Our intuition is, if $g_{\boldsymbol{\theta}_1}$ can realistically simulate the noise, then enforcing $g_{\boldsymbol{\theta}_2}(\mathbf{Z}) + g_{\boldsymbol{\theta}_1}(\boldsymbol{r})$ to mimic $\mathbf{Z}$ would result in learning a reasonable initial *denoiser* $g_{\boldsymbol{\theta}_2}$. One important detail regarding $g_{\boldsymbol{\theta}_2}$ is its final activation *must* be the sigmoid function for stable training. The third loss (7), which

---

[1]We assume $g_{\boldsymbol{\theta}_2}$ implicitly has the cropping step for $\mathbf{Z}$ such that the dimension of $g_{\boldsymbol{\theta}_2}(\mathbf{Z})$ and $g_{\boldsymbol{\theta}_1}(\boldsymbol{r})$ match.

resembles the cycle loss in (Zhu et al., 2017), imposes the encoder-decoder structure between $g_{\theta_2}$ and $g_{\theta_3}$, hence, helps $g_{\theta_2}$ to compress the most redundant part of $\mathbf{Z}$, *i.e.*, the noise, and carry out the initial denoising. Once the losses are defined, training the generators and critics are done in an alternating manner, as in the training of W-GAN (Arjovsky et al., 2017), to approximately solve

$$\min_{\theta_1,\theta_2,\theta_3} \max_{w_1,w_2} \Big[ \alpha\mathcal{L}_n(\theta_1,w_1) + \beta\mathcal{L}_{\mathbf{Z}}(\theta_1,\theta_2,w_2) + \gamma\mathcal{L}_{\text{cyc}}(\theta_2,\theta_3) \Big], \qquad (8)$$

in which $(\alpha,\beta,\gamma)$ are hyperparameters to control the trade-offs between the loss functions. The pseudo algorithm for training a generative model is given in Algorithm 1. There are a couple of subtle points for training with the overall objective (8), and we describe the full details on model architectures and hyperparameters in the S.M.

---

**Algorithm 1** Training a generative model, all experiments in this paper used the defaults values, $n_{critic}=5$, $n_{epoch}=30$, m = 64, $\alpha_g = 4e^{-4}$, $\alpha_{critic} = 5e^{-5}$, $\alpha = 5$, $\beta = 1$, $\gamma = 10$

1: **Require** $\mathcal{D}, \lambda$
2: $\mathcal{N} \leftarrow \text{NoisePatchExtraction}(\mathcal{D}, \lambda)$
3: **for** $ep_{GAN} \leftarrow 1, n_{epoch}$ **do**
4: $\quad$ Sample $\{n^{(i)}\}_{i=1}^m \sim \mathcal{N}$, $\{r^{(i)}\}_{i=1}^m \sim N(0,I)$, $\{Z^{(i)}\}_{i=1}^m \sim \mathcal{D}$
5: $\quad$ **for** $ep_{critic} \leftarrow 1, n_{critic}$ **do**
6: $\qquad$ $g_{w_1} \leftarrow \nabla_{w_1}[\mathcal{L}_n(\theta_1,w_1)]$, $g_{w_2} \leftarrow \nabla_{w_2}[\mathcal{L}_Z(\theta_1,\theta_2,w_2)]$
7: $\qquad$ $w_1 \leftarrow Clip(w_1 + \alpha_{critic} \cdot Adam(w_1,g_{w_1}), -c, c)$
8: $\qquad$ $w_2 \leftarrow Clip(w_2 + \alpha_{critic} \cdot Adam(w_2,g_{w_2}), -c, c)$
9: $\quad$ **end for**
10: $\quad$ $g_{\theta_1}, g_{\theta_2}, g_{\theta_3} \leftarrow \nabla_{\theta_1,\theta_2,\theta_3}[\alpha\mathcal{L}_n(\theta_1,w_1) + \beta\mathcal{L}_Z(\theta_1,\theta_2,w_2) + \gamma\mathcal{L}_{\text{cyc}}(\theta_2,\theta_3)]$
11: $\quad$ $\theta_1 \leftarrow \theta_1 - \alpha_g \cdot Adam(\theta_1,g_{\theta_1})$, $\theta_2 \leftarrow \theta_2 - \alpha_g \cdot Adam(\theta_2,g_{\theta_2})$, $\theta_3 \leftarrow \theta_3 - \alpha_g \cdot Adam(\theta_3,g_{\theta_3})$
12: **end for**
13: **return** $\theta_1, \theta_2$

---

### 4.3 ITERATIVE GAN2GAN TRAINING OF A DENOISER

With our generative model, we then carry out the iterative Noisy N2N training as described in Section 3, with the *generated* noisy images. Namely, given each $\mathbf{Z}^{(i)} \in \mathcal{D}$, we generate the pair

$$(\hat{\mathbf{Z}}_{11}^{(i)}, \hat{\mathbf{Z}}_{12}^{(i)}) \triangleq (g_{\theta_2}(\mathbf{Z}^{(i)}) + g_{\theta_1}(r_{11}^{(i)}), g_{\theta_2}(\mathbf{Z}^{(i)}) + g_{\theta_1}(r_{12}^{(i)})), \qquad (9)$$

in which $r_{11}^{(i)}, r_{12}^{(i)} \in \mathbb{R}^{128}$ are i.i.d. $\sim \mathcal{N}(\mathbf{0}, I)$. In contrast to the ideal case in Section 3, each generated image in (9) is a noise-corrupted version of $g_{\theta_2}(\mathbf{Z}^{(i)})$, in which the corruption is done by the *simulated* noise $g_{\theta_1}(r)$. Denoting the set of such pairs as $\hat{\mathcal{D}}_1 = \{(\hat{\mathbf{Z}}_{11}^{(i)}, \hat{\mathbf{Z}}_{12}^{(i)})\}_{i=1}^n$, a denoiser $\hat{X}_{\phi}(\mathbf{Z})$ is trained by minimizing $\mathcal{L}_{\text{G2G}}(\phi, \hat{\mathcal{D}}_1) \triangleq \frac{1}{n}\sum_{i=1}^n (\hat{\mathbf{Z}}_{11}^{(i)} - \hat{X}_{\phi}(\hat{\mathbf{Z}}_{12}^{(i)}))^2$. In $\mathcal{L}_{\text{G2G}}(\cdot)$, we only use the generated noisy images and do *not* use the actual observed $\mathbf{Z}^{(i)}$, hence, we dubbed our training as GAN2GAN (G2G) training. Now, denoting the learned denoiser as G2G$_1$ (with parameter $\phi_1$), we can iterate the G2G training. For the $j$-th iteration (with $j \geq 2$), we generate

$$(\hat{\mathbf{Z}}_{j1}^{(i)}, \hat{\mathbf{Z}}_{j2}^{(i)}) \triangleq (\hat{X}_{\phi_{j-1}}(\mathbf{Z}^{(i)}) + g_{\theta_1}(r_{j1}^{(i)}), \hat{X}_{\phi_{j-1}}(\mathbf{Z}^{(i)}) + g_{\theta_1}(r_{j2}^{(i)})), \qquad (10)$$

for each $\mathbf{Z}^{(i)}$ and denote the resulting set of the pairs as $\hat{\mathcal{D}}_j$. Note in (10), we *update* the noisy estimate of the clean image with the output of G2G$_{j-1}$. Then, the new denoiser G2G$_j$ is obtained by computing $\phi_j \triangleq \arg\min_{\phi} \mathcal{L}_{\text{G2G}}(\phi, \hat{\mathcal{D}}_j)$, where the minimization is done via warm-starting from $\phi_{j-1}$. In our experiments, we show the sequence, G2G$_{j \geq 1}$, successively refines the denoising quality and significantly improves the initial noisy estimate, similarly as in Figure 1. Moreover, we identify the benefit of the iterative G2G training becomes greater when noise is more sophisticated; *i.e.*, for synthetic noise, the performance of G2G$_{j \geq 1}$ converges after 1~3 iterations, whereas for the real-world microscopy noise, the performance keeps increasing until larger number of iterations.

## 5 EXPERIMENTAL RESULTS

### 5.1 DATA AND EXPERIMENTAL SETTINGS

**Data & training details** In synthetic noise experiments, we always used the noisy training images from BSD400 (Martin et al., 2001). For evaluation, we used the standard BSD68 (Roth & Black,

2009) as a test set. For real-noise experiment, we experimented on two data sets: the WF set in the microscopy image datasets in (Zhang et al., 2019) and the reconstructed CT dataset. For both datasets, we trained/tested on each (Avg = $n$) and each dose level, respectively, which corresponds to different noise levels. For the generative model training, the patch size used for $\mathcal{D}$ and $\mathcal{N}$ was $96 \times 96$, and $n$ and $N$ were set to $20,000$ (BSD) and $40,000$ (microscopy), respectively. For the iterative G2G training, the patch size for $\mathcal{D}$ was $120 \times 120$ and $n = 20,500$, and in every mini-batch, we generated new noisy pairs with $g_{\boldsymbol{\theta}_1}$ as in the noise augmentation of (Zhang et al., 2017). The architecture of G2G$_j$ was set to 17-layer DnCNN in (Zhang et al., 2017). We put full details on training, model architectures and hyperparameters as well as the software platforms in the S.M.

**Baselines** The baselines were BM3D (Dabov et al., 2007), DnCNN-B (Zhang et al., 2018), N2N (Lehtinen et al., 2018), and N2V (Krull et al., 2019). We reproduced and trained DnCNN-B, N2N and N2V using the publicly available source codes on the *exactly* same training data as our iterative G2G training. For DnCNN-B and N2N, which use either clean targets or two independent noisy image copies, we used 20-layers DnCNN model with composite additive white Gaussian noise with $\sigma \in [0, 55]$. N2V considers the same setting as ours and uses the *exact* same architecture as G2G$_j$; more details on N2V are also given in the S.M. We could not compare with the scheme in (Laine et al., 2019), since their code cannot run beyond white Gaussian noise case in our experiments and they had an unfair advantage: they *newly* generate noisy images by corrupting given clean images for *every* mini-batch whereas we assume the given noisy images are fixed once for all. It is known that such noise augmentation significantly can increase the performance, and their code could not run in our setting in which the noisy images are fixed once given. As an upper bound, we implemented N2C(Eq.(4)), denoting a 17-layer DnCNN trained with clean target images in BSD400 and their noisy counterpart, which is corrupted by our $g_{\boldsymbol{\theta}_1}$ learned with (4).

## 5.2 DENOISING RESULTS ON SYNTHETIC NOISE

**White Gaussian noise** Table 2 shows the results on BSD68 corrupted by white Gaussian noise with different $\sigma$'s. Several variations of our G2G, $g_{\boldsymbol{\theta}_2}$ and the G2G iterates, G2G$_{j \geq 1}$, are shown for two different training data versions for learning the generative model. Firstly, we clearly observe the

Table 2: Results on *BSD68/Gaussian*. Boldface denotes algorithms that only use single noisy images. Red and blue denotes the highest and second highest result among those algorithms, respectively.

| PSNR/SSIM | Baselines | | | | G2G variation | | | | Upper Bound |
|---|---|---|---|---|---|---|---|---|---|
| | BM3D | DnCNN-B | N2N | N2V | $g_{\theta_2}$ | G2G$_1$ | G2G$_2$ | G2G$_3$ | N2C(Eq.(4)) |
| $\sigma = 15$ | **31.07/0.8717** | 31.44/0.8836 | 31.20/0.8745 | **29.48/0.8199** | **25.94/0.7519** | **30.98/0.8552** | **32.51/0.8827** | **31.45/0.8825** | 31.64/0.8870 |
| $\sigma = 25$ | **28.56/0.8013** | 28.92/0.8137 | 28.74/0.8041 | **26.97/0.7083** | **24.16/0.6630** | **28.23/0.7669** | **28.82/0.8056** | **28.96/0.8080** | 29.11/0.8189 |
| $\sigma = 30$ | **27.78/0.7727** | 28.06/0.7812 | 27.91/0.7720 | **26.38/0.6657** | **23.43/0.5967** | **27.58/0.7413** | **27.99/0.7783** | **28.03/0.7759** | 28.28/0.7890 |
| $\sigma = 50$ | **25.60/0.6866** | 25.78/0.6721 | 25.71/0.6712 | **24.30/0.5765** | **20.58/0.4482** | **25.08/0.6215** | **25.55/0.6639** | **25.78/0.6749** | 26.03/0.6951 |

iterative G2G training is *very* effective; namely, it significantly improves the initial noisy estimate $g_{\boldsymbol{\theta}_2}$, particularly when the quality of the initial estimate is not good enough. This result confirms the result of Figure 1 indeed carries over to the blind denoising setting with our method. Secondly, we note G2G$_1$ already considerably outperforms N2V, which is trained with the *exact* same model architecture and dataset. Finally, the performance of G2G$_3$ is *very* strong; it outperforms BM3D, which knows true $\sigma$, and even sometimes outperforms the blindly trained DnCNN-B and N2N, which is trained with the same BSD400 dataset, but with more information. This somewhat counter-intuitive result is possible since our G2G$_j$ accurately learns the correct noise level in the image, while DnCNN-B and N2N are trained with the composite noise levels, $\sigma \in [0, 55]$.

**Mixture and correlated noise** Table 3 shows the results on mixture and correlated noise beyond white Gaussian. Note our G2G$_j$ does not assume any distributional or correlation structure of the noise, hence, it can still run as long as the assumption on the noise holds. In the table, the G2G results

Table 3: Results on *BSD68/Mixture & Correlated noise*. The boldface and colored texts are as before.

| PSNR/SSIM | | | Baselines | | | | G2G variation | | | | Upper bound |
|---|---|---|---|---|---|---|---|---|---|---|---|
| | | | BM3D | DnCNN-B | N2N | N2V | $g_{\theta_2}$ | G2G$_1$ | G2G$_2$ | G2G$_3$ | N2C(Eq.(4)) |
| Mixture noise | Case A | $s = 15$ | **41.44/0.9822** | 39.62/0.9749 | 40.59/0.9860 | **33.53/0.9368** | **31.85/0.9522** | **42.35/0.9876** | **42.56/0.9888** | **42.49/0.9885** | 42.92/0.9843 |
| | | $s = 25$ | **37.97/0.9647** | 37.23/0.9616 | 37.39/0.9737 | **31.62/0.9057** | **32.73/0.9478** | **39.13/0.9761** | **39.64/0.9800** | **39.72/0.9807** | 40.42/0.9843 |
| | Case B | $s = 30$ | **30.12/0.8549** | 30.58/0.8655 | 30.36/0.8559 | **28.10/0.7543** | **27.55/0.7728** | **29.05/0.8199** | **30.32/0.8456** | **30.49/0.8538** | 30.78/0.8685 |
| | | $s = 50$ | **29.27/0.8190** | 30.20/0.8547 | 30.20/0.8547 | **28.22/0.7755** | **27.36/0.7712** | **29.78/0.8345** | **30.04/0.8392** | **30.00/0.8417** | 30.39/0.8574 |
| Correlated noise | | $\sigma = 15$ | **29.84/0.8504** | 30.84/0.9011 | 30.69/0.9223 | **28.80/0.8367** | **28.13/0.8370** | **30.73/0.8889** | **31.09/0.8949** | **31.26/0.8954** | 31.60/0.9075 |
| | | $\sigma = 25$ | **26.69/0.7544** | 27.39/0.8257 | 27.32/0.8594 | **26.11/0.7348** | **25.68/0.7607** | **27.80/0.8130** | **28.01/0.8271** | **28.00/0.8447** | 28.42/0.8376 |

are for (BSD) as specified above. Moreover, DnCNN-B and N2N are still blindly trained with the *mismatched* white Gaussian noise. For mixture noise, we tested with two cases. Case A corresponds to the same setting as given in (Chen et al., 2018), *i.e.*, $70\% \sim \mathcal{N}(0, 0.1^2)$, $20\% \sim \mathcal{N}(0, 1)$, and

$10\% \sim \text{Unif}[-s, s]$ which means the random variable that is uniformly distributed between $[-s, s]$ with $s = 15, 25$. For case B, we tested with larger variances, *i.e.*, 70% Gaussian $N(0, 15^2)$, 20% Gaussian $N(0, 25^2)$, and 10% Uniform $[-s, s]$ with $s = 30, 50$. For correlated noise, we generated the following noise for each $\ell$-th pixel,

$$N_\ell = \eta M_\ell + (1 - \eta)\Big(\frac{1}{\sqrt{|\mathcal{NB}_\ell|}} \sum_{m \in \mathcal{NB}_\ell} M_m\Big), \ \ \ell = 1, 2, \dots$$

in which $\{M_\ell\}$ are white Gaussian $\mathcal{N}(0, \sigma^2)$, $\mathcal{NB}_\ell$ is the $k \times k$ neighborhood patch except for the pixel $\ell$, and $\eta$ is a mixture parameter. We set $\eta = 1/\sqrt{2}$ such that the marginal distribution of $N_\ell$ is also $\mathcal{N}(0, \sigma^2)$ and set $k = 16$. Note in this case, $N_\ell$ has a spatial correlation, and we tested with $\sigma = 15, 25$. From the table, we first note that DnCNN-B and N2N suffer from serious performance degradation for both mixture and correlated noises due to noise mismatch, and the conventional BM3D outperforms them for some cases (*e.g.*, Case A for mixture noise). However, we note our G2G$_2$ can still denoise very well after just two iterations and outperforms all the baselines for all noise types. Note N2V seriously suffers and is *not* comparable to ours. Finally, N2C(Eq.(4)) is a sound upper bound for all noise types, confirming the correctness of the extraction rule (4).

## 5.3 DENOISING RESULTS ON REAL NOISE

We also test our method on the real-world noise. While some popular real noise is known to have source-dependent characteristics, there are also cases in which the noise is source-independent and pixel-wise correlated, which satisfies the assumption of our method. We tested on two such datasets, the Wide-Focal (WF) set in the microscopy image dataset (Zhang et al., 2019) and a Reconstructed CT dataset. A more detailed description and analysis on these two datasets are in S.M. The WF

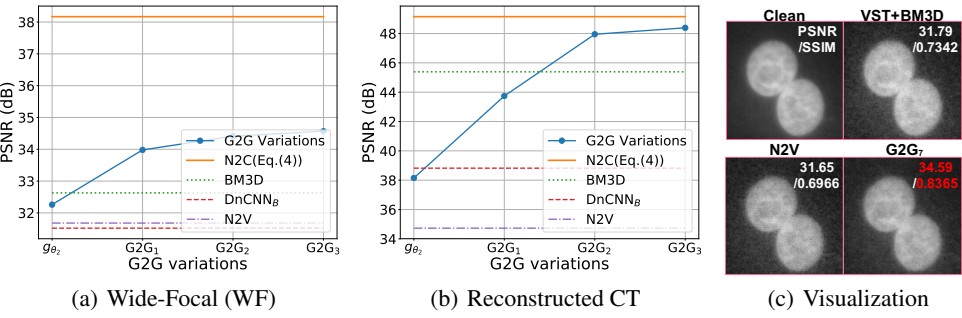

(a) Wide-Focal (WF)      (b) Reconstructed CT      (c) Visualization

Figure 4: Results on real microscopy image denoising on WF and medical image denoising.

and Reconstructed CT data has 5 sets (Avg = 1, 2, 4, 8, 16) and 4 sets (Dose=25, 50, 75, 100) with different noise levels, respectively. We did *not* exploit the fact that the images are multiple noisy measurements of a clean image, which enables employing N2N, but treated them as noisy images of distinct clean images. Figure 4(a) and 4(b) shows the PSNR of all methods for each dataset, respectively, averaged over all sets. The baselines were DnCNN-B, BM3D and N2V. We note BM3D estimated noise $\sigma$ using the method in Chen et al. (2015). We iterated until G2G$_3$ and N2C(Eq.(4)) was an upper bound for each set. We clearly observe that the performance of G2G$_j$ significantly improves (over $g_{\theta_2}$) as the iteration continues. In results, G2G$_3$ becomes significantly better than DnCNN-B and N2V as well as BM3D, still one of the strongest baselines for real-world noise denoising when no clean target images are available, for both datasets. We report more detailed experimental results (including SSIM) on both datasets in S.M. Moreover, the inference time for BM3D is about 4.5∼5.0 seconds per image since a noise estimation has to be done for each image separately, whereas that for G2G$_j$ is only 4 ms (on GPU), which is another significant advantage of our method. Figure 4(c) shows the visualizations on the WF, and we give more examples in the S.M.

## 5.4 ABLATION STUDY

**Noise patch extraction** Here, we evaluate the effect of the noisy patch extraction rules (3) and (4) in the final denoising performance. Figure 5 compares the PSNR of N2C(GCBD Eq.(3)), a re-implementation of (Chen et al., 2018), N2C(Ours Eq.(4)) and the best G2G, for each dataset.

We note neither source code nor training data of (Chen et al., 2018) is publicly available, and the PSNR in (Chen et al., 2018) could not be reproduced (with the exact same $\eta$ and $\gamma$ as in (Chen et al., 2018)). From the figure, we clearly observe the significant gap between N2C(Our Eq.(4)) and N2C(GCBD Eq.(3)), particularly when the noise is not white Gaussian. Moreover, our *pure* unsupervised G2G with (4) even outperforms N2C(GCBD Eq.(3)) that utilizes the clean target images, confirming the quality difference shown in Figure 2(b) significantly affects learning noise and a denoiser.

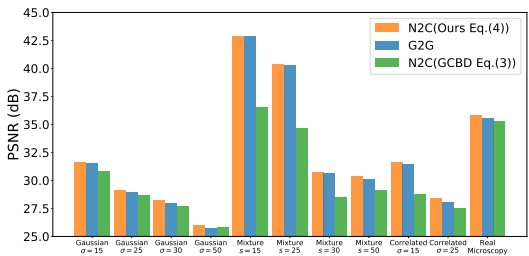

Figure 5: Effect of noise patch extraction rule.

**Generative model and iterative G2G training** Figure 6(a) shows the PSNRs of $g_{\boldsymbol{\theta}_2}$ on BSD68/Gaussian($\sigma = 25$) trained with three variations; "No $\mathcal{L}_{\mathbf{Z}}$" for no $f_{\mathbf{w}_2}$, "No $\mathcal{L}_{\text{cyc}}$" for no (7) and $g_{\boldsymbol{\theta}_3}$, and "No sigmoid" for no sigmoid activation at the output layer of $g_{\boldsymbol{\theta}_2}$. We confirm that our proposed architecture achieves the highest PSNR for $g_{\boldsymbol{\theta}_2}$, the sigmoid activation and $f_{\mathbf{w}_2}$ are essential, and the cylce loss (7) is also important. Achieving a decent PSNR for $g_{\boldsymbol{\theta}_2}$ is beneficial for saving the number of G2G iterations and achieving high final PSNR. More detailed analyses on the generative model architecture are in the S.M. Figure 6(b) and 6(c) show the effect of the quality of

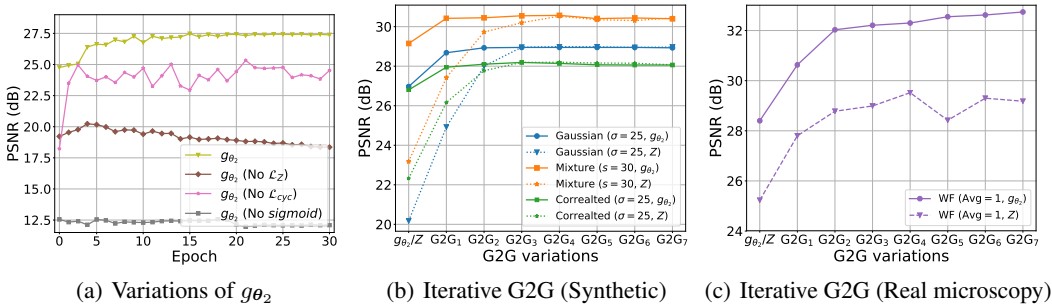

(a) Variations of $g_{\boldsymbol{\theta}_2}$     (b) Iterative G2G (Synthetic)     (c) Iterative G2G (Real microscopy)

Figure 6: Ablation studies. (b) and (c) compare the performances between starting from $g_{\boldsymbol{\theta}_2}$ and $\mathbf{Z}$.

the initial estimate for the iterative G2G training. From Figure 1, one may ask whether $g_{\boldsymbol{\theta}_2}$ is indeed necessary, since even when $\sigma_0 \approx \sigma$, the iterating the Noisy N2N can mostly achieve the upper bound. Hence, for samples of synthetic and real microscopy data, we evaluate how G2G$_j$ performs when the iteration simply starts with $\mathbf{Z}$. Figure 6(b) shows a somewhat surprising result that for synthetic noises, starting from $\mathbf{Z}$ achieves essentially the same performance as starting from $g_{\boldsymbol{\theta}_2}$ with a couple more G2G iterations. However, for real microscopy noise case in Figure 6(c), WF(Avg= 1) in which starting from $\mathbf{Z}$ achieves far lower performance than starting from $g_{\boldsymbol{\theta}_2}$, justifying our generative model for attaining the initial noisy estimate.

## 6 CONCLUDING REMARK

Motivated by a novel observation on Noisy N2N, we proposed a novel GAN2GAN method, which can tackle the challenging blind image denoising problem solely with single noisy images. Our method showed impressive denoising performance that even sometimes outperform the methods with more information as well as VST+BM3D for real noise denoising. As a future work, we plan to extend our framework to more explicitly handle the source-dependent real-world noise.

ACKNOWLEDGMENT

This work was supported in part by NRF Mid-Career Research Program [NRF-2021R1A2C2007884] and IITP grant [No.2019- 0-01396, Development of framework for analyzing, detecting, mitigating of bias in AI model and training data], funded by the Korean government (MSIT).

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
