# OpenReview forum: "GAN2GAN: Generative Noise Learning for Blind Denoising with Single Noisy Images"
_ICLR.cc/2021/Conference — ICLR 2021 Poster_

### Official Review · AnonReviewer3 · 2020-10-17
**The idea looks reasonable, but I have several concerns.**

**Rating:** 7
**Confidence:** 4

**Review:**

This paper proposes a framework to train a network to remove noises, which are zero-mean, additive and independent of the clean image, with only noisy images and without knowing the noise statistics. They mathematically prove that a network, which is trained from pairs of images generated by adding simulated noises into the noisy image, can remove noises from the input noisy image. The proposed framework can remove the noises from the input noisy image. Then, it adds simulated noises into the denoised image to generate a pair of images and train the next network to further remove noises and it does this process iteratively. The proposed framework follows GCBD and utilizes flat textureless regions to train a network to simulate noises and proposes a wavelet based method to effectively distinguish flat regions from the ones that contain high-frequency repeating patterns. The experimental results show that the proposed method has good performance under simulated Gaussian noises as well as WT and CT datasets.

I have several concerns and suggestions for this paper:

(1) One assumption of the proposed method is the noises and signal are independent which is not true for most of the scenarios. The raw images captured by the camera contain both Poisson noises. I am wondering if the proposed method can deal with Poisson noises or not. Also, the final displayed images processed by ISP contain signal dependent noises with spatial correlation. Can the proposed framework remove this kind of noises?

(2) In Sec. 3.3, why does the framework add simulated noises into the network 'ground truth' Z_j1^i. Will it be better if we directly treat X_phi_{j-1}(Z^i) in Eq. (10) as the 'ground truth'?

(3) As claimed above Theorem 1, f_Noisy N2N (Z, y) gives a better estimate of X than X' for a sufficiently large σ_0^2. What the results will be if σ_0^2 is not large enough? That is to say, can the proposed framework remove small noises from the input images?

(4) Similarly, according to Theorem 1, the iterative process in Sec. 3.3 will converge if y_0<y<1.  Can the network still effectively remove noises if y<y_0 which means the residual noises are not significant in the images?

(5) g_theta1 generates noises with given random vector r. How does g_theta3 not need a random vector to add simulated noises into g_theta2(Z)?

(6) More analyses are needed in Sec. 4.4 for ablation study. Especially, why is sigmoid so important?

Some descriptions are not clear and confusing:

(7) I cannot understand the relationship between N2C with Eq. 4.

(8) What is Unif[−s, s] in Sec. 4.2?

(9) It is not clear the described dataset for Table 3 for training or testing.

(10) For a fair comparison,  do the two N2C networks have the same network structure?

---

### Official Review · AnonReviewer2 · 2020-10-25
**This paper proposes a blind image denoising method based on a self-supervised learning approach.**

**Rating:** 4
**Confidence:** 3

**Review:**

The proposed method uses the generative learning method to simulate the noise and synthesize noisy image pairs to train the proposed network. Experimental results show the effectiveness of the proposed method.

Learning denoised network from noisy images has been developed by Krull et al., 2019; Batson
& Royer, 2019; Laine et al., 2019. The main difference is the use of generative learning. One possible clarification is whether the aforementioned methods using generative learning can generate comparable results or not. If so, the contribution of the paper is limited.

The motivation of using the proposed network design is not clear. In addition, using more generators will lead to larger capacity models than existing methods. It is not clear whether the performance gains are due to use such larger capacity models or not.

For the real noise images, why do the authors evaluate the proposed method on the microscopy and medical images? How about the results on the real-world natural noisy images? In addition, the proposed results still contain significant noise residual as shown in Figure 4.

---

### Official Review · AnonReviewer4 · 2020-10-26
**Solid work with sound technical approach, but better with analysis on other well known approaches.**

**Rating:** 7
**Confidence:** 3

**Review:**

This paper addresses a challenging task of blind image denoising where a single noisy image is provided with assumption that it is zero mean, additive and independent from the original image content. This is mostly the real-world scenario. Different from the recent N2N training, the authors propose a GAN2GAN based method since this blind setting cannot be trained by N2N. N2N or deterministic training needs explicit or implicit knowledge of clean image in order to be trained whereas the GAN2GAN method does not, leading to more realistic and efficient training. This method first attempts to simulate noise given the noisy image, generate rough and noisy estimates of the clean image, and iteratively train a denoiser with synthetic noisy pairs from the generator. For blind denoising, this work produces impressive results for synthetic and real world blind denoising.

This work provides a sound approach to denoising given only the noisy image.
The experiments were done with extensive datasets and baselines.
Although the paper is solid, I am not sure how this work compares to well known algorithms for blind image denoising:
- Residual Dense Network for Image Restoration
- High-Quality Self-Supervised Deep Image Denoising
- CycleISP: Real Image Restoration via Improved Data Synthesis
- Real Image Denoising with Feature Attention
just to name a few.
If the authors can provide comparisons or discussion, analysis on these algorithms and comparison, it would be helpful.

---

### Official Review · AnonReviewer1 · 2020-10-29
**interesting idea of iterative denoiser training**

**Rating:** 7
**Confidence:** 5

**Review:**


This paper proposed a new method for blind image denoising using a "GAN2GAN" network. Different from previous work noise2noise, the proposed method only needs single noisy images to train the network, without noisy pairs. Given a noisy dataset, a GAN generator is trained using the real noisy but smooth patches. With the noise generator, a pair of generated noise samples are added on the noisy image content to train a denoiser. This is trained iteratively, so that the image content is better and better after cleaned by the trained denoiser. The final results on synthetic and real data showed improvements over the baselines.

Overall, this paper is very interesting to read. It solves a harder problem than noise2noise when the pairs are not available. The idea of refining the noisy image iteratively is reasonable, which makes the image content for training progressively better. The results showed significant improvements over its competitor noise2void.

There are still some areas that it could improve:

1. It is unclear the purpose of the cycle loss. Although it improves the PSNR, the motivation is not clear.

2. The idea of generating synthetic noise for image denoising has been studied already. This should be discussed.

Abdelhamed, Abdelrahman, Marcus A. Brubaker, and Michael S. Brown. "Noise flow: Noise modeling with conditional normalizing flows." Proceedings of the IEEE International Conference on Computer Vision. 2019.

3. It would be great to have natural images in the experiments to validate the idea. In particular, showing that the proposed method can reduce and noise and reveal more image content and texture.

Abdelhamed, Abdelrahman, Stephen Lin, and Michael S. Brown. "A high-quality denoising dataset for smartphone cameras." Proceedings of the IEEE Conference on Computer Vision and Pattern Recognition. 2018.

Plotz, Tobias, and Stefan Roth. "Benchmarking denoising algorithms with real photographs." Proceedings of the IEEE conference on computer vision and pattern recognition. 2017.

4. It is good to have a table or flowchart about the iteratively training, to make the paper easier to read.

---

### Decision · Program_Chairs · 2021-01-07
**Final Decision**

**Decision:**

Accept (Poster)

**Comment:**

Summary of discussion: Three reviewers rated the paper Good (7) while Reviewer2 disagreed. R2's criticism was focussed on how this work is placed within existing/related literature, and no technical problem was identified. The authors have addressed some of R2's comments/concerns, R2 has not participated in the discussion.

Novelty and contributions: Overall the reviews seem consistent with an incremental paper which is technically valid, improves the state of the art on a reasonably difficult task. However, it does not appear from the reviews that the paper substantially advances our understanding of machine learning more broadly beyond this specific application.

Experiments: There is some disagreement among reviewers on the adequacy of the experiments, with at least two reviewers calling for experiments involving 'natural photos'. I believe the author's responses adequately address these concerns: they pointed out that the key selling point of their paper is the ability to model structured noise which is less relevant in natural photos.

On the balance of things, I think this paper should be accepted, but I wouldn't argue if it did not make the cut due to its narrow scope. For this reason, I recommended poster presentation.